# Time Synchronization and Space Registration of Roadside LiDAR and Camera

**Chuan Wang [1], Shijie Liu [2,3,*], Xiaoyan Wang [4,*] and Xiaowei Lan [3,5]**

1   Shandong High-Speed Group Co., Ltd., Jinan 250098, China
2   School of Microelectronics, Shandong University, Jinan 250101, China
3   Suzhou Research Institute, Shandong University, Suzhou 215000, China
4   Shandong Academy of Transportation Science, Jinan 250357, China
5   School of Transportation, Lanzhou Jiaotong University, Lanzhou 730070, China
\*   Correspondence: 202032465@mail.sdu.edu.cn (S.L.); sonorra@163.com (X.W.);
    Tel.: +86-132-1044-1585 (S.L.); +86-139-6409-6114 (X.W.)

**Abstract:** The sensing system consisting of Light Detection and Ranging (LiDAR) and a camera provides complementary information about the surrounding environment. To take full advantage of multi-source data provided by different sensors, an accurate fusion of multi-source sensor information is needed. Time synchronization and space registration are the key technologies that affect the fusion accuracy of multi-source sensors. Due to the difference in data acquisition frequency and deviation in startup time between LiDAR and the camera, asynchronous data acquisition between LiDAR and camera is easy to occur, which has a significant influence on subsequent data fusion. Therefore, a time synchronization method of multi-source sensors based on frequency self-matching is developed in this paper. Without changing the sensor frequency, the sensor data are processed to obtain the same number of data frames and set the same ID number, so that the LiDAR and camera data correspond one by one. Finally, data frames are merged into new data packets to realize time synchronization between LiDAR and camera. Based on time synchronization, to achieve spatial synchronization, a nonlinear optimization algorithm of joint calibration parameters is used, which can effectively reduce the reprojection error in the process of sensor spatial registration. The accuracy of the proposed time synchronization method is 99.86% and the space registration accuracy is 99.79%, which is better than the calibration method of the Matlab calibration toolbox.

**Keywords:** camera; frequency self-matching; joint calibration; LiDAR; space synchronization

## 1. Introduction

Sensors are widely used in the field of traffic. Light Detection and Ranging (LiDAR) and a camera can provide real-time perception information of the surrounding environment [1,2], which can provide powerful data support for vehicle–road cooperation technology. To improve the target fusion effect [3–8], time synchronization and space registration of sensor data information are needed to achieve multi-source sensors' robust information perception ability in different environments.

As the main component of the visual perception system, the camera can provide abundant color and image information at a low cost, and become the indispensable hardware base for the comprehensive perception of complex road conditions [9]. However, the camera is greatly affected by illumination changes, and its performance stability is reduced in dark conditions. With the rapid development of the 3D laser industry, LiDAR can obtain high angles and speed resolution, provide rich 3D data information, is not affected by lighting conditions, and has been widely studied and applied in the field of traffic [10]. However, LiDAR cannot provide the color information of the target. It can only rely on the three-dimensional size information to track the target, which is prone to target-matching errors.

Therefore, the fusion technology can realize the complementary advantages of LiDAR and camera [11,12], which can effectively improve the performance and efficiency of target detection [13,14]. However, due to the different frequencies and spatial location of data acquisition between LiDAR and the camera [15–17], time synchronization and space registration need to be solved. Otherwise, these easily lead to data fusion deviation [18–20]. For example, the time deviation of data acquisition between LiDAR and the camera can lead to an obvious position deviation of the same target, affecting object calibration and detection accuracy [21,22]. Therefore, solving the problems of time synchronization and space registration between LiDAR and camera is a critical technical link to improving the accuracy of data fusion [23–25].

To solve these problems, this paper proposes a simple and efficient sensor time synchronization method and adopts a space registration method based on a nonlinear optimization algorithm. These two methods solve the problems of time synchronization and space registration between LiDAR and camera data, effectively reducing the amount of data after time synchronization of the multi-source sensor. When the LiDAR and camera positions are different, the space registration of multi-source sensor data is realized, and the ideal data fusion effect is obtained. Finally, the robustness of the proposed method is verified by comparing it with relevant spatial synchronization methods.

The remaining parts of the paper are structured as follows: Section 2 presents a brief review of relevant work. Section 3 introduces the data acquisition as well as the principle of time synchronization and the space registration of sensors. Section 4 introduces the experiment and data processing. Section 5 analyzes the comparison between this study and other methods. Section 6 shows the conclusion and related future works.

## 2. Related Work

Due to the frequent application of LiDAR and camera in recent years, fusion technology has been continuously developed. As a prerequisite for data fusion of multi-source sensors, time synchronization and space registration have been studied by some researchers [26–29] (such as hardware synchronization, clock synchronization, network synchronization, etc.) to solve the problem of extrinsic parameter calibration in different sensor modes. Meanwhile, the necessity of sensor time synchronization and space registration is reflected in the research and development of many intelligent systems and devices. Therefore, time synchronization and space registration between multi-source sensors have attracted much attention and research.

Firstly, time synchronization methods are mainly divided into hardware synchronization and software synchronization. For example, the internal clocks of different sensors can synchronize time based on the same GPS reference clock, so that data information of multiple sensors can be matched and relevant data processing can be realized [30–32]. The hardware synchronization method based on the GPS reference clock is mainly applied to sensors with related interfaces. At present, most projects in the field of intelligent transportation are based on the Ubuntu robot operating system (ROS). The ROS system contains the simplest method of time soft synchronization. Sensors connected to the ROS system perform time matching when data arrive at the computer and need to subscribe to the topics of different sensors. By viewing timestamp data headers from topics published by different sensors [33,34], which are time synchronized to receive different topics, synchronized functions for data processing are finally. However, this method has a large time deviation and low efficiency, and the system is easy to crash when processing data from multiple sensors. Therefore, Furgale et al. proposed a new framework to jointly estimate the time deviation between different sensor measurements and the spatial displacement between them. It is realized through a continuous time batch processing estimation, and the time deviation is seamlessly combined in the strict theoretical framework of maximum likelihood estimation [35]. However, although this method can accurately calculate and eliminate the time deviation, it has a large amount of data and takes a long time. Zhang et al. proposed a simple self-calibration method for the internal time synchronization of

MEMS (micro-electro-mechanical System) LiDAR. This method can automatically calculate whether the sensor has performed time synchronization, without any manual participation. Finally, an actual experiment on MEMS LiDAR was carried out to verify the effectiveness of this method [36]. This method is only applicable to the internal calibration of LiDAR and fails to achieve time synchronization in multiple sensors, such as camera and LiDAR.

At present, there are many methods of calibrating LiDAR and camera with extrinsic parameters [37–39]. The calibration method is mainly divided into two parts: one is based on dynamic calibration and the other is based on point, line, and plane calibration. The process of dynamic calibration is mainly used to calibrate the trajectory of the LiDAR and camera sensors. Meanwhile, the relationship between image attitude estimation, point cloud attitude estimation, and vehicle attitude information are mainly judged by similarity evaluation between tracks. Based on the point, line, and plane method, 3D point clouds and 2D images are matched directly for calibration.

Zhang et al. earlier proposed an extrinsic calibration theory consisting of a camera and a two-dimensional laser rangefinder, pointing out that the angle between the calibration plate plane and the laser scanning plane can affect the calibration accuracy [40]. Xiang et al. proposed a joint calibration method based on the correspondence principle of the distance from the sensor origin to the calibration plane [41]. However, the influence of the calibration plate attitude on the calibration result is not studied, which is not conducive to improving the accuracy of the calibration result. Chai et al. proposed a 3D–2D corresponding feature method for LiDAR and camera calibration [42], and then performed rigid body transformation calculations to obtain more stable calibration results. Lyu et al. used an interactive LiDAR camera calibration toolbox to calculate the transformation of intrinsic and extrinsic parameters. The corner of the board can be automatically detected through the LiDAR data frame. The board used here refers to the two-dimensional code calibration board. Meanwhile, the toolbox uses genetic algorithms to estimate and support multiple transformations [43]. Pusztai et al. studied the calibration of a vision system consisting of LiDAR and RGB camera sensors and proposed a new method for calibration using cardboard boxes of known sizes [44]. At present, time synchronization and space registration between sensors are often applied in automatic driving, but there is still insufficient research on the static detection of urban road vehicles and pedestrians by roadside sensors [45,46]. Moreover, the dynamic calibration method cannot achieve high stability [47], and the sensor time synchronization requirements are very high and need to introduce a time offset. Adding an offset will reduce the calibration accuracy [48].

However, the time synchronization and space registration of sensors are mainly reflected in autonomous driving, and the research on the time synchronization and spatial matching of roadside LiDAR and cameras lacks objective generalization ability. Especially in the aspect of time synchronization, there are not many detailed theoretical derivations and objective evaluation methods. At present, the calibration method of Matlab is the most commonly used and relatively stable calibration method [49]. In this paper, we propose a time synchronization method of frequency self-matching for roadside sensors. Based on the time synchronization and the relative position differences between roadside LiDAR and camera, a nonlinear optimization algorithm is introduced to effectively reduce the reprojection error in the process of spatial synchronization. Compared with the existing Matlab toolbox calibration methods, the feasibility of the proposed method is proven.

## 3. Data Collection and Methods

### 3.1. Data Collection

In this experiment, RS-LiDAR-32 and a camera were used for data acquisition (as shown in Figure 1), and the single-lens camera of the Jieruiweitong brand was adopted with a resolution of 1080P. The RS-LIDAR-32 has 32 laser transceiver modules for 360-degree scanning, with a scanning speed of 5 Hz, 10 Hz, and 20 Hz. Table 1 describes the parameters of the RS-LIDAR-32; the data are from the 32-line mechanical LiDAR technical manual on the official website of Shenzhen Sagitar Juchuang Technology Co., Ltd.

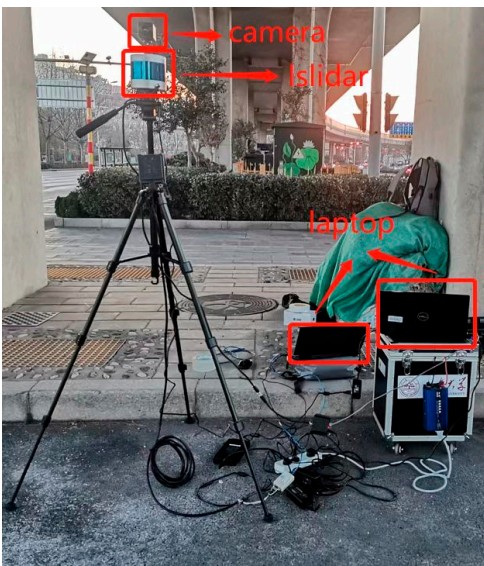

**Figure 1.** LiDAR and camera data acquisition system.

**Table 1.** The LiDAR parameters.

| Indicator | Value |
|---|---|
| Laser beams | 32 |
| Scan FOV | $40° \times 360°$ |
| Vertical angle resolution | $0.33°$ |
| Rotation rate | 300/600/1200 (r/min) |
| Laser wavelength | 905 nm |
| Vertical field of view | $-16°\sim+15°$ |
| Operating temperature | $-20\sim60\ °C$ |
| Single echo data rate | 650,000 points/s |
| Measuring range | 100 m~200 m |
| Communication Interface | PPS/UDP |

### 3.2. Time Synchronization

Due to the different frequencies of data acquisition, there is time asynchronism between roadside LiDAR and camera data acquisition. To effectively fuse the collected data, time synchronization is required, and there are two methods of processing: (1) Time synchronization based on LiDAR data, and (2) time synchronization based on camera data. As the frequency of the LiDAR sensor is less than that of the camera, to reduce the calculation amount of the method in this paper and prevent the over-fitting phenomenon in the subsequent neural network training process, we choose the LiDAR data as the reference for time synchronization. LiDAR is used as the core sensor for time synchronization. Every frame of LiDAR data is received, the current data are used as the starting point for time interpolation, and two frames of data information of the camera, before and after this time point, are obtained.

It is assumed that the interval of point cloud data of each frame of LiDAR is $T_1$ seconds, and the gap of an image of each camera frame is $T_2$ seconds. Because the data collection frequency of LiDAR is less than the collection frequency of camera data, $T_2$ is less than $T_1$ (namely, $T_2 < T_1$), therefor the threshold should be $T_2/2$. The two sensors are set to start working at time zero; this means that the camera and the LiDAR use the same device for data acquisition. After the same time, the LiDAR obtains $m$ frames per point cloud, and the camera obtains $n_m$ ($n_m = mT_1/T_2$) frames per image.

The next frame image is numbered as the $n_{m+1}\left(n_{m+1} = [mT_1/T_2] + 1 = \left[\frac{mT_1}{T_2} + 1\right]\right)$, where all square brackets in this section indicate that they integers. When $mT_1/T_2 -$

$[mT_1/T_2] < T_2/2$, the point cloud of frame $m$ and the image of frame $n_m$ form a set of point pairs. $mT_1/T_2 - [mT_1/T_2]$ represents the difference between the time used by the $n_m$ frame image and the time used by the corresponding $m$ frame LiDAR data. If the difference is less than the set threshold, the image data of this frame can be retained; otherwise, they will be eliminated. When $\left[\frac{mT_1}{T_2}+1\right] - mT_1/T_2 < T_2/2$, the point cloud of frame $m$ and the image of frame $n_{m+1}$ form a set of point pairs. $\left[\frac{mT_1}{T_2}+1\right] - mT_1/T_2$ represents the difference between the time used by the $n_{m+1}$ frame image and the time used by the corresponding $m$ frame LiDAR data. If the difference is less than the set threshold, the image data of this frame can be retained; otherwise, they will be eliminated. The remaining image information that does not satisfy the above two inequalities is eliminated. Eventually, the LiDAR and camera data are combined into new packets. The time synchronization of the information collected by the two sensors is realized through the operation process.

*3.3. Space Registration*

To optimize the reprojection error of the camera and LiDAR in the process of space synchronization, a nonlinear optimization algorithm with joint calibration parameters is adopted. However, in nonlinear joint calibration, the alignment of data between the LiDAR and camera needs to be considered. Therefore, based on the above time synchronization, the space synchronization between the LiDAR and camera is studied. It is basic work to calibrate the camera's intrinsic parameters before joint calibration. The main process is the conversion between the pixel coordinate system, image coordinate system, LiDAR coordinate system, camera coordinate system, and world coordinate system. The camera's intrinsic parameters are calibrated through the Zhang calibration method [50].

To determine the calibration plate plane, the point cloud coordinates of the calibration plate plane should be extracted first. The calibration plane is the reference plane in the process of spatial synchronization between the camera and LiDAR, namely the calibration plate. It is helpful to determine the relationship between a point in the three-dimensional space of the LiDAR and the corresponding geometric position in the image so that the spatial synchronization between the camera and the LiDAR is more accurate. KD-tree, as a data structure, is used to represent the set of points in k-dimensional space [51]. The KD-tree method in the PCL library is used to select the point cloud around a specified point. Due to the complexity of the test environment, there is no pre-processing of point cloud data. Therefore, according to the selected point cloud data, the random sampling consensus algorithm (RANSAC) is used to calculate the plane equation of the calibration plate. Let the fitting plane equation be $ax + by + cz + s = 0$ where $a$, $b$, $c$, and $s$ are used as unknown fitting parameters, and four points near the center of the plane plate are selected to calculate the plane equation. Other 3D point cloud coordinates are substituted into Formula (1) to calculate the distance between 3D points and the fitting plane.

$$D = \frac{|ax + by + cz + s|}{\sqrt{a^2 + b^2 + c^2}}, \tag{1}$$

The threshold is set to U. If the distance between other 3D points is greater than U, they are deleted. Finally, 3D points within the threshold are left. The number of iterations is set as 2000, and the points in U in each iteration are compared. The fitting plane with the most points is the real plane.

After finding the optimal plane, calibration optimization is needed, and the main purpose of optimization is to reduce the reprojection error in the 3D–2D point pair projection

process. Finally, the space synchronization of LiDAR and camera is realized. The process of projecting the 3D point cloud onto the image is as follows:

$$Z \begin{bmatrix} u \\ v \\ 1 \end{bmatrix} = \begin{bmatrix} f_x & 0 & u_0 & 0 \\ 0 & f_y & v_0 & 0 \\ 0 & 0 & 1 & 0 \end{bmatrix} \begin{bmatrix} R & T \\ \overrightarrow{0} & 1 \end{bmatrix} \begin{bmatrix} X \\ Y \\ Z \\ 1 \end{bmatrix},$$ (2)

where $l = [X, Y, Z]^T$ represents the 3D point cloud coordinates of LiDAR. $R$ stands for the rotation matrix $R = (r_{11}, r_{12}, r_{13}, \ldots, r_{31}, r_{32}, r_{33})$. $T$ stands for the translation matrix $T = (t_x, t_y, t_z)$. $K = (f_x, f_y, u_0, v_0)$ represents the camera's internal parameter. $Z$ represents distance depth information. $(v, u)$ represents the pixel coordinate points of the actual projection. Among them, the parameters that need to be optimized are $R$ and $T$. In the optimization process, nonlinear function optimization is introduced to minimize the reprojection error and obtain more accurate values of $R$ and $T$. Based on time synchronization between camera and LiDAR data, parameters were optimized by using the recursive optimization method (LM) improved by the gradient descent method and the Gauss–Newton method. The main advantage of an LM algorithm is to ensure fast convergence while ensuring decline. Since the position of the camera and LiDAR relative to the calibration plate is unknown, the camera and LiDAR, under different attitudes, are calibrated by moving the calibration plate in the process of space synchronization, so a least-squares problem is constructed using the inclination angle and azimuth angle of the camera and LiDAR relative to the calibration plate as the error source. Therefore, the global loss function between LiDAR and camera can be defined as:

$$W_{\text{lf\_lc}} = \sum_{i=1}^{n} \left( \| \varphi_{li} - \varphi_{ci} \|^2 + \| \theta_{li} - \theta_{ci} \|^2 \right),$$ (3)

In Formula (3), $\varphi_{li}$, $\varphi_{ci}$, respectively, represent the inclination angles of the LiDAR and camera with respect to the calibration plate, and $\theta_{li}$, $\theta_{ci}$, respectively, represent the azimuth angles of the LiDAR and camera with respect to the calibration plate. Based on the above nonlinear optimization method, the reprojection error between the LiDAR and the camera should be calculated According to the principle of reprojection error, let $T_{lc} = \begin{bmatrix} R & T \\ \overrightarrow{0} & 1 \end{bmatrix}$, $P_{li}$ be the point projected onto the camera coordinate system by the 3D point cloud. $P_{ci}$ represents 3D points projected onto the image coordinate system by the camera's internal matrix $K$. According to Formula (2), the reprojection error of the checkerboard calibration board can be expressed as:

$$\Im = \sum_{i=1}^{n} \left\| P_{ci} - \frac{1}{Z} K T_{lc} P_{li} \right\|^2,$$ (4)

In Formula (4), the reprojection error in the space synchronization process can be obtained, where $\Im$ represents the reprojection error in the sensor calibration process.

## 4. Experimental Analysis

### 4.1. Time Synchronization Verification Method

Based on the time synchronization principle mentioned above, a flat road surface was selected for the time synchronization verification test of the multi-source sensor. A certain brand of car was used as the vehicle for the verification test. We placed three triangular cones in a straight line every 5 m along the road direction. The three triangular cones were numbered successively, and the first triangular cone was placed at the center of the left front wheel of the car, taking the center point as the starting point, as shown in Figure 2. We started the car and drove at a constant speed. The constant speed of the car was 40 km/h. Through the driver's operation on the vehicle, the vehicle entered the fixed speed cruise mode, to ensure that the vehicle traveled at a uniform speed. The car passed the second and

third triangular cones in turn. When the center of the front wheel of the vehicle overlapped with the second and third triangular cones (as shown in Figures 3 and 4), the point cloud data at each triangular cone was obtained by LiDAR, as shown in Figure 5. According to Formula (5), the coordinates $(\overline{x_i}, \overline{y_i}, \overline{z_i})$ of each point cloud center were be obtained, and then the distance $(L_{i1})$ between the point cloud center of any triangular cone and the point cloud center at the starting point were calculated according to Formula (6).

$$\overline{x_i} = \frac{1}{n}\sum_{j=1}^{n} x_{ij}; \ \overline{y_i} = \frac{1}{n}\sum_{j=1}^{n} y_{ij}; \ \overline{z_i} = \frac{1}{n}\sum_{j=1}^{n} z_{ij},$$
$$i = (1,2,3); \ j = (1,2,\dots,n) \tag{5}$$

where, $\overline{x_i}$, $\overline{y_i}$, and $\overline{z_i}$ represent the location coordinates of the LiDAR point cloud center at the front wheel of the vehicle.

$$L_{i1} = \sqrt{\left(\overline{x_i} - \overline{x_1}\right)^2 + \left(\overline{y_i} - \overline{x_1}\right)^2 + \left(\overline{z_i} - \overline{z_1}\right)^2} \tag{6}$$

where $L_{i1}$ represents the distance between the LiDAR point cloud center at the front wheel of the vehicle and the starting point.

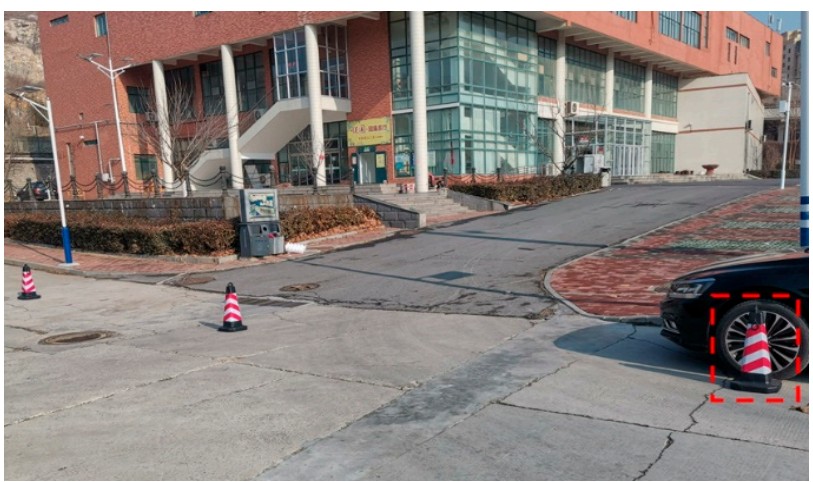

**Figure 2.** The starting point of the test.

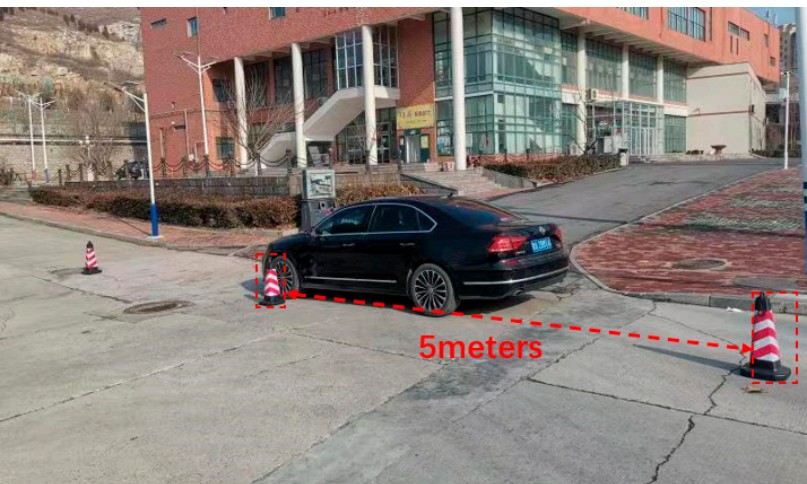

**Figure 3.** Vehicle driving distance of 5 m.

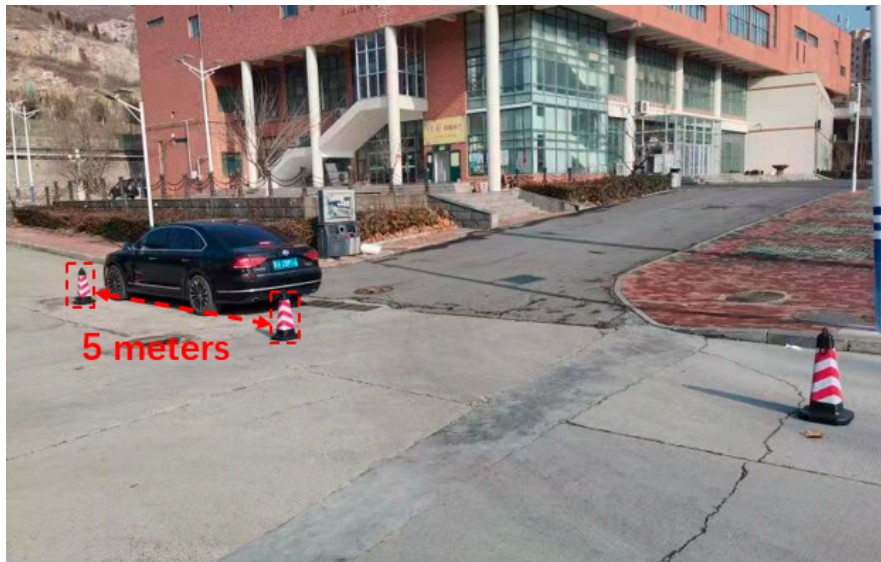

**Figure 4.** Vehicle driving distance of 10 m.

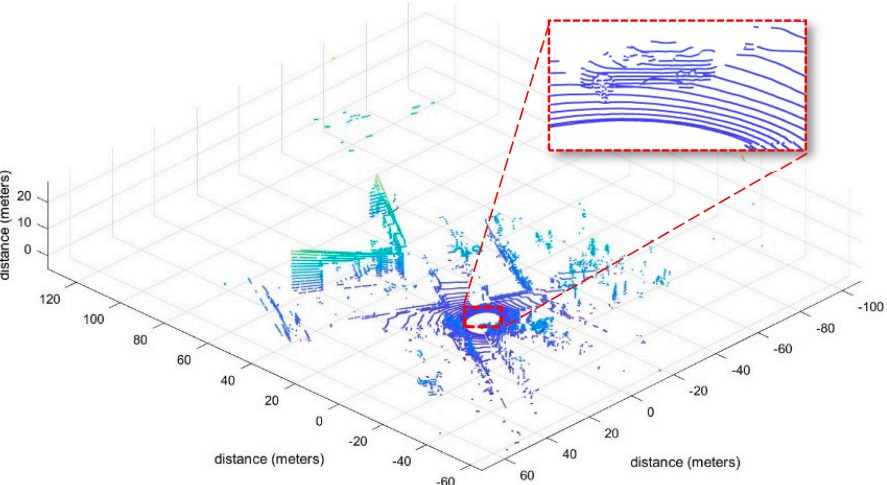

**Figure 5.** LiDAR point cloud diagram.

With the left front wheel used as the target, the vehicle was driven from the first triangular cone to the third, while the LiDAR and camera were tested. This was repeated for a total of 10 tests. Ten groups of data after time synchronization were obtained in the test and compared with the vehicle position distance before time synchronization by the sensor.

The results are shown in Figures 6 and 7. Figure 6 shows the position deviation when the vehicle moved 5 m, while Figure 7 shows the comparison of position deviation when the vehicle moved 10 m. As can be seen from the two figures, the data deviation after sensor realizes time synchronization is less than that without time synchronization. After time synchronization, the accuracy of moving 5 m and 10 m can reach 99.86 percent and 99.49 percent, respectively.

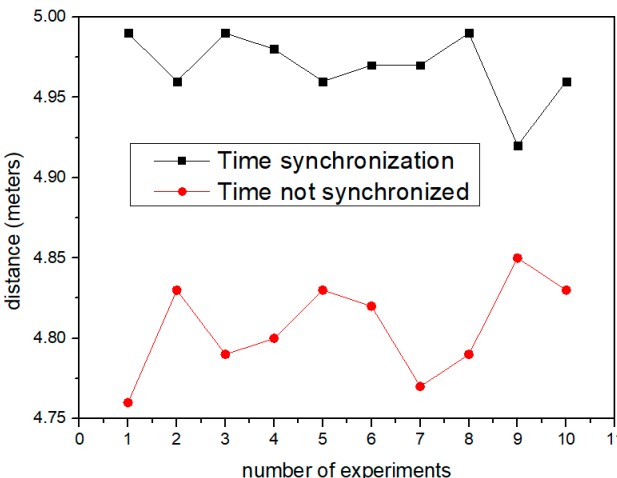

**Figure 6.** Comparison of distance errors before and after synchronization when the vehicle drives a distance of 5 m.

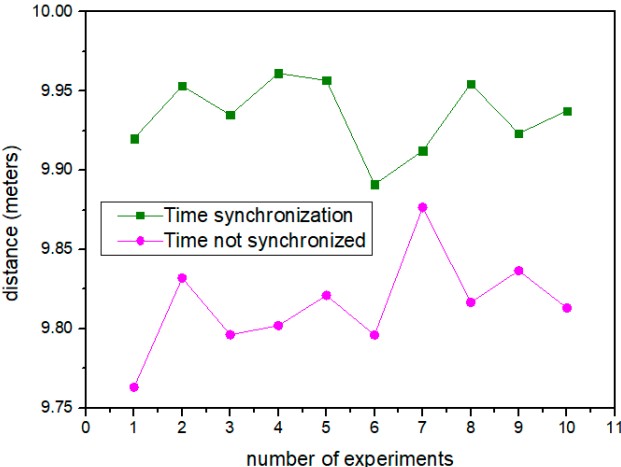

**Figure 7.** Comparison of distance errors before and after synchronization when the vehicle dives a distance of 10 m.

### 4.2. Space Registration Verification Method

Based on the above two theoretical methods, the road of the author's campus was selected as the test site. During the experiment, corner points inside the calibration board were identified according to the position changes of the calibration board, as shown in Figure 8. Finally, the point cloud was effectively projected onto the calibration board. The black point cloud part represents the background point where the LiDAR scans the surrounding environment, and the red point represents the point cloud scanned onto the calibration board. During the moving process of the calibration board, the inclination and azimuth of the calibration board at different times were be recorded, as shown in Figure 9.

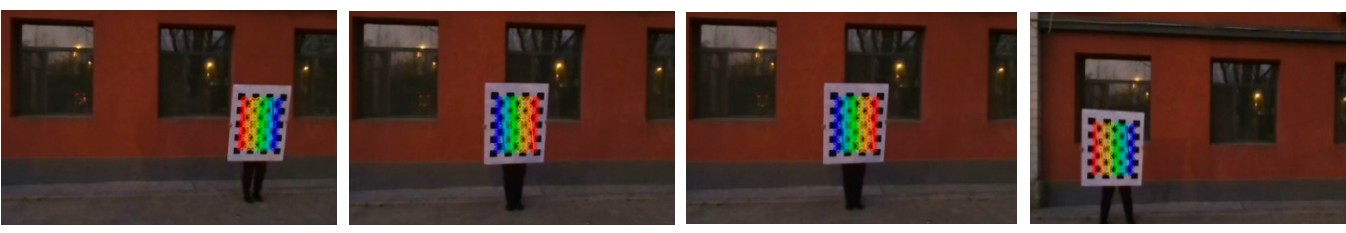

**Figure 8.** Identification of the corner points of the calibration board.

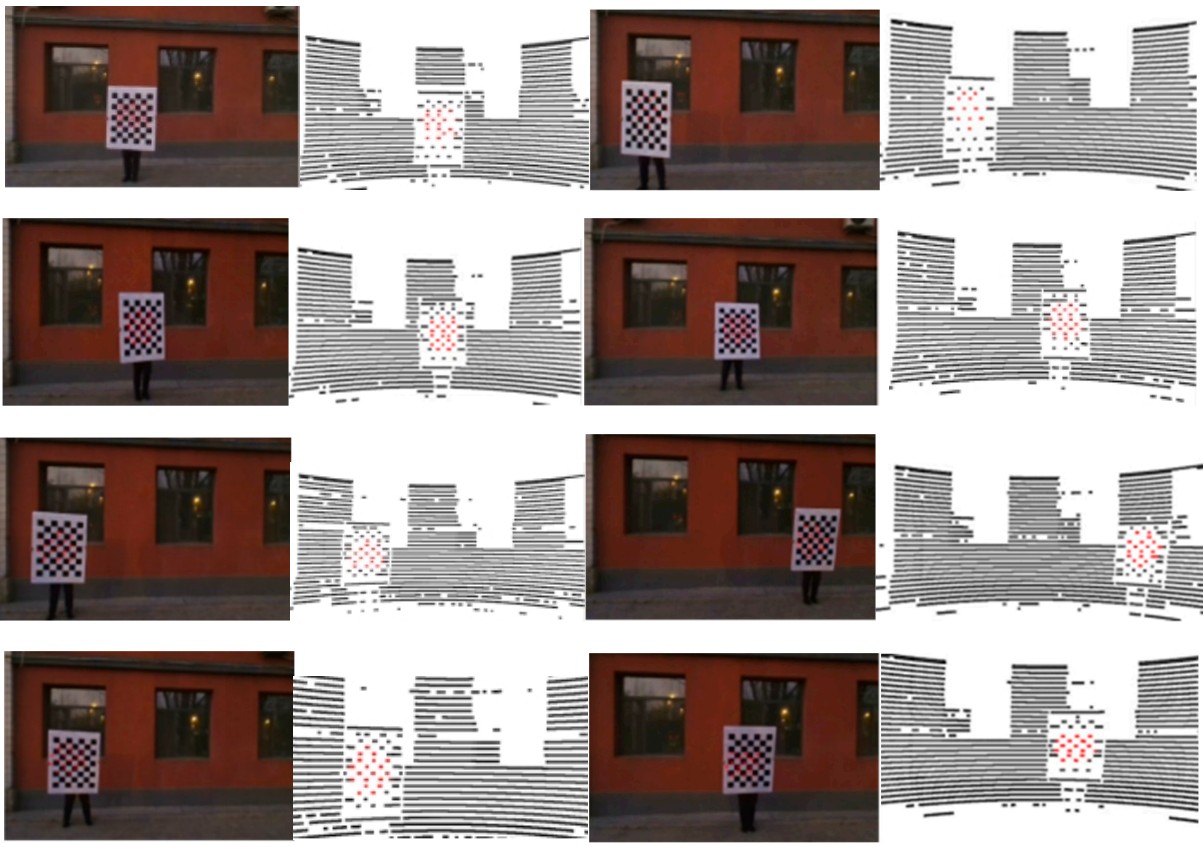

**Figure 9.** Point cloud calibration plate reprojection.

Through the calibration of the camera's internal parameters and the calibration of the external parameters between the LiDAR and the camera, the rotation parameter matrix and shift parameter matrix are, respectively, obtained as follows:

$$R = \begin{bmatrix} -6.5577526781824158e-02 & 4.9700067859074659e-02 & 9.9660899616448506e-01 \\ -9.9778758283539681e-01 & -1.4208891757146347e-02 & -6.4946492857811400e-02 \\ 1.0932864248457241e-02 & -9.9866311168974153e-01 & 5.0521894555605629e-02 \end{bmatrix}$$

$$T^{-1} = \begin{bmatrix} -1.8226658771695058e-02 \\ -8.1717580020885579e-03 \\ 2.4798673724301969e-02 \end{bmatrix}$$

where $R$ is the rotation matrix between the LiDAR and the camera, which can be interpreted as the projection of the LiDAR coordinate system to the camera coordinate system through the transformation matrix $R$. $T^{-1}$ is the transformation matrix from camera to LiDAR, and $T^{-1}$ is composed of rotation matrix $R$ and translation vector $T$.

## 5. Results and Discussions

Based on the application of time synchronization and space registration methods of LiDAR and camera data in this paper, after time synchronization and space registration, the fusion experiment was carried out under the condition that the relative position of LiDAR and camera remained unchanged. The ideal fusion of LiDAR and camera was obtained, as shown in Figure 10. In the process of joint calibration of LiDAR and camera, different relative positions of LiDAR and camera were selected for a space registration experiment under the common view, and the comparison error of the obtained results is shown in Table 2. Table 2 shows that the reprojection error changes when the LiDAR and the camera are in different relative positions. When the height difference between the LiDAR and the

camera is less than 20 cm and the horizontal distance is less than 150 cm, the fusion effect is the best.

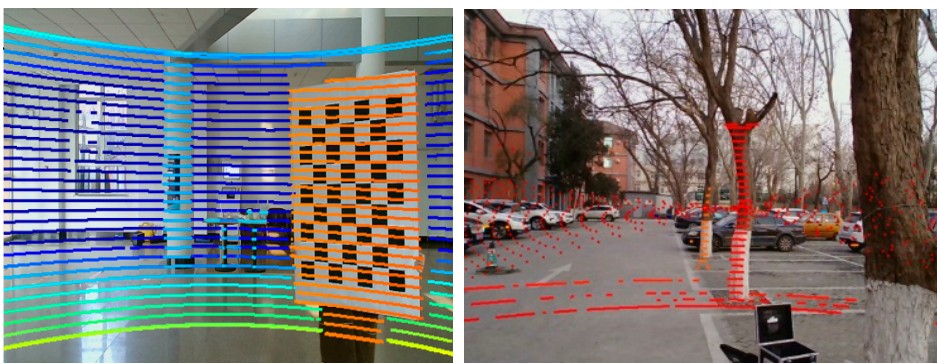

**Figure 10.** Effect of LiDAR and camera data fusion.

**Table 2.** Reprojection errors of LiDAR and camera at different distances.

| Height (cm) | Horizontal Distance (cm) | Reprojection Error (Pixel) |
|---|---|---|
| 10 | 50 | 0.159981 |
| 10 | 100 | 0.166632 |
| 10 | 150 | 0.263361 |
| 10 | 200 | 0.339633 |
| 20 | 50 | 0.169532 |
| 20 | 100 | 0.176923 |
| 20 | 150 | 0.294632 |
| 20 | 200 | 0.369654 |
| 30 | 50 | 0.219987 |
| 30 | 100 | 0.321463 |
| 30 | 150 | 0.322134 |
| 30 | 200 | 0.329786 |

As the Matlab calibration toolbox is one of the advanced and convenient calibration technologies, the proposed LiDAR and camera time synchronization and space registration method was compared with the Matlab toolbox method based on Zhou et al. [52], and field experiments were carried out to verify the comparison data results, as shown in Figure 11, which provides a comparison and reference scheme for time synchronization, space registration, and data fusion of multi-source sensors. The comparison test used the same instruments and equipment for data acquisition, and was performed in the same experimental environment.

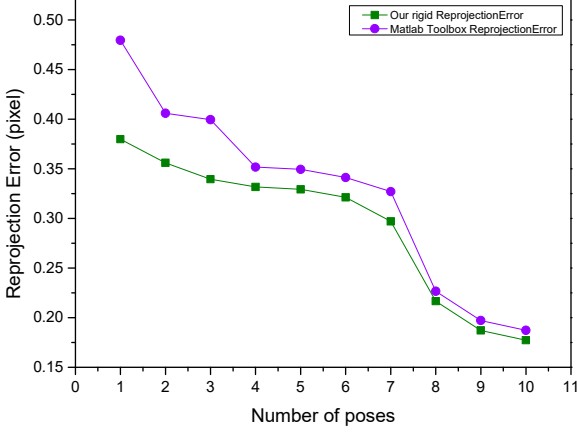

**Figure 11.** Reprojection error comparison.

## 6. Conclusions

To solve the problem of the cumbersome calibration process between LiDAR and camera, a time synchronization method based on self-frequency and a space synchronization method based on a nonlinear optimization algorithm are proposed in this study, which can effectively realize the time synchronization between LiDAR and camera data. Based on time synchronization, a nonlinear optimization algorithm of joint calibration parameters is proposed, which can effectively reduce the reprojection error in the process of space registration of sensors. Compared with the existing work, the randomness and complexity of the experimental scenarios of time synchronization and space registration in this paper can be applied to most environments and provide accurate results.

The main innovations of the method established in this study are as follows: 1. Through visualization experiments, the time synchronization accuracy of LiDAR and camera data can reach 99.86%. Compared with non-time synchronization, the accuracy of data after time synchronization is improved by 9.63%. Meanwhile, the time synchronization process is simplified and the time synchronization efficiency is improved. 2. Based on the time synchronization of roadside sensor data information, the method can accurately and efficiently realize sensor space registration and adapt to different complex environments.

This method provides a practical and effective solution for the data fusion of vehicle–road cooperative multi-source sensor equipment. For example, it can be used to solve the influence of relative position on data fusion in the installation process of different roadside sensors. Meanwhile, the method is also suitable for the traffic scene in a complicated urban environment.

There are also some limitations to this paper. For instance, only one kind of LiDAR was used for data collection in the experiment, and the experiment under rain and snow and other bad weather conditions was not carried out, which is the content that needs to be made up in the next research work. At present, we have made a plan to carry out experiments with different types of LiDAR. Meanwhile, the next step will be synchronizing LiDAR, millimeter-wave radar, and cameras in time and space.

**Author Contributions:** Conceptualization, S.L. and C.W.; methodology, S.L.; software, X.L.; validation, X.W., S.L. and C.W.; formal analysis, S.L.; investigation, C.W.; resources, X.L.; data curation, X.W.; writing—original draft preparation, S.L. and C.W.; writing—review and editing, X.L.; visualization, C.W.; supervision, C.W.; project administration, X.W.; funding acquisition, C.W. All authors have read and agreed to the published version of the manuscript.

**Funding:** This research was funded partly by the National Natural Science Foundation of China, grant number 52002224, partly by the National Natural Science Foundation of Jiangsu Province, grant number BK20200226, partly by the Program of Science and Technology of Suzhou, grant number SYG202033, and partly by the Key Research and Development Program of Shandong Province, grant number 2020CXG010118.

**Data Availability Statement:** The data used to support the findings of this study are available from the corresponding author upon request.

**Conflicts of Interest:** The authors declare no conflict of interest.

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
