# Peer review of "Time Synchronization and Space Registration of Roadside LiDAR and Camera"

_electronics, doi:10.3390/electronics12030537_

Round 1

Reviewer 1 Report

Dear Authors,

Thank you very much for reviewing your paper and answering my questions. Please add the content of your answers to your paper so that other readers who potentially ask themselves the same questions as I do can also benefit from it. For example, it is important to know the speed of the vehicle in order to estimate the temporal deviation between the camera and LIDAR and transfer it to a distance.

In particular, I would like to suggest that you highlight the assumptions you have made, for example regarding the common timeline for camera and LiDAR. Only in this way can your paper contribute to the scientific discussion.

Please also explain what the "calibration plate plane" is (L193) and how to work with it.

Please add the approach from Matlab to the state of the art. Since you only compare your results with this one approach, the statement in L27 that your approach is better than the state of the art is formulated too generally. Please clearly elaborate where your approach differs from the state of the art in science and technology.

Yours sincerely

Reviewer 2 Report

The authors present a testbed (experimental study) for determining space-time optimization of a LiDaR in a street topology for vehicular application. Pixel error is calculated as an error vector. The paper is well written and interesting in its subject matter and overall presentation. My singular observation lies in the 3.1 and 3.2 sections. The authors are kindly requested to provide sources/references or other explanation for the metrics in Table 1 and the equations provided in 3.2. This is the numerical basis for the further testbed carried out by the authors, therefore it is important to clarify the source for these equations and values.

Round 2

Reviewer 1 Report

Dear Authors,

Thank you very much for the new version of the paper. I see some improvements but would like to suggest to continue in this direction. To help others to benefit from your research, you should consider the following points:

You need to improve the English language. It makes it difficult to read and understand the paper.

Include the explanations/answers you gave me in the paper. They are helpful. In parallel, please delete the new line (204/205) about the calibration plane. This is not helpful. Instead, explain to the reader that the plane is a real (physical) plate that is placed in front of the measurement systems at different angles and positions to perform the geometric calibration.

L126 Explain which board you are referring to.

Add a reference to the statement in lines 141/142.

L199-202 The different coordinate systems are not clearly defined. Please add an explanation of each of them (this explanation could also be a sketch).

L234 The calibration plate itself cannot be calibrated.

Chapter 4 Please indicate the speed of the vehicle, at least approximately.

L289 10 meters, not 5.

L330 The statement that the reprojection error remains unchanged is not clear since the error in table 2 changes.

Yours sincerely.
